# Programmable catalysis by support polarization: elucidating and breaking scaling relations

Seongjoo Jung [1], Cristina Pizzolitto[2], Pierdomenico Biasi [2], Paul J. Dauenhauer[1,3] & Turan Birol [1] ✉

The Sabatier principle and the scaling relations have been widely used to search for and screen new catalysts in the field of catalysis. However, these powerful tools can also serve as limitations of catalyst control and breakthrough. To overcome this challenge, this work proposes an efficient method of studying catalyst control by support polarization from first-principles. The results demonstrate that the properties of catalysts are determined by support polarization, irrespective of the magnitude of spontaneous polarization of support. The approach enables elucidating the scaling relations between binding energies at various polarization values of support. Moreover, we observe the breakdown of scaling relations for the surface controlled by support polarization. By studying the surface electronic structure and decomposing the induced charge into contributions from different atoms and orbitals, we identify the inherent structural property of the interface that leads to the breaking of the scaling relations. Specifically, the displacements of the underlying oxide support impose its symmetry on the catalyst, causing the scaling relations between different adsorption sites to break.

The historical process of catalyst design has focused on permanent modification of the electronic structure of active sites through optimization of composition, crystal structure, or use of supports[1]. This approach has achieved major advances in catalysis but confronts the fundamental limitations of static catalytic sites called the Sabatier principle[2]. The catalytic sites with peak performance exhibit intermediate binding energies of key intermediates to balance the rates of two or more elementary reaction steps.

There is continuing focus on the catalysis design that goes beyond the Sabatier limit[3]. A recent microkinetic modeling study identified the linear scaling relations (LSRs), especially those involving reaction intermediates, such as transition state scaling (TSS) relations[4] and Brønsted-Evans-Polanyi (BEP) relations[5] as the origins of the Sabatier volcano[6] (Fig. 1a). Methods that break the LSRs between different interaction energies of adsorbate and surface with external perturbation include activation of reagents or catalyst surfaces by plasma, strain, or magnetic fields[7–9].

Another strategy is dynamic programmable catalysis, where surfaces oscillate between two or more states with distinct binding energies of key reaction intermediates[10]. By switching between surface electronic states at a frequency comparable to the catalytic turnover, the reaction is no longer limited by the rate-determining step (RDS) of either state and can take advantage of fast independent reaction steps. Dynamic programmable catalysts have been recently demonstrated by methods that use light or electron energy in faradaic reactions to alter the reaction energetics. Gopeesingh et al. demonstrated over one order of magnitude (20x) increased turnover frequency by applying -100 Hz oscillating potential to the Pt working electrode in electrochemical formic acid decomposition reaction[11]. Lim et al. reported a 553% increase in the rate of reaction using -0.1 Hz oscillating potentials

[1]Department of Chemical Engineering & Materials Science, University of Minnesota, 421 Washington Ave. SE, Minneapolis, MN 55455, USA. [2]Casale SA, Via Giulio Pocobelli 6, CH-6900 Lugano, Switzerland. [3]Center for Programmable Energy Catalysis (CPEC), University of Minnesota, 421 Washington Ave. SE, Minneapolis, MN 55455, USA. ✉e-mail: tbirol@umn.edu

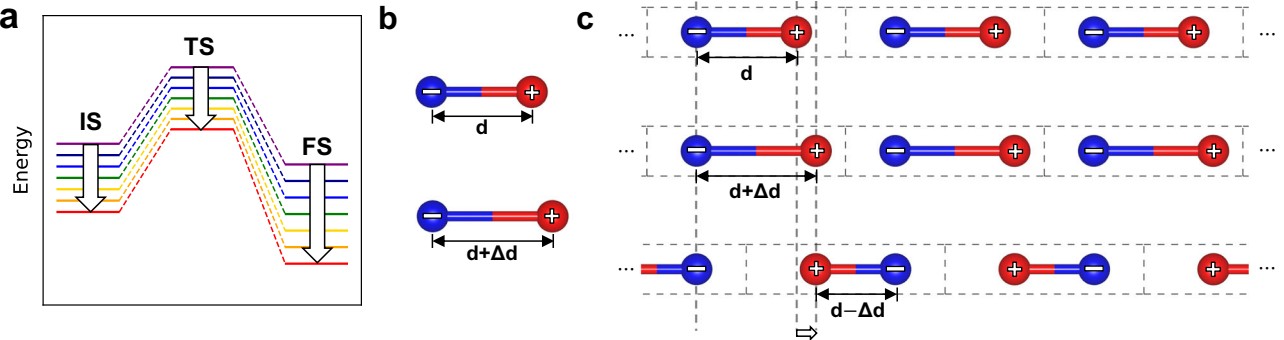

**Fig. 1 | Illustration of transition state scaling with a positive slope and polarization in solids. a** Usually, a surface modification that makes the initial state (IS) more stable by stronger surface-adsorbate interaction also makes transition state (TS) and final state (FS) more stable, disallowing significant changes in the reaction kinetics. **b** For a molecule, displacement of charge results in a straightforward change in dipole moment. **c** Displacement of a positively charged particle in a 1-D solid. For crystalline solids, displacement of charge can be interpreted in both the increase and decrease of the absolute value of dipole moment, depending on the choice of the unit cell.

to promote non-Faradaic ethylene hydrogenation on a Pd/C electrode[12]. Qi et al. showed ~50% increase in the photocatalytic rate of methanol decomposition on Pt nanoparticles with pulsed illumination up to 3.5 kHz[13].

A method of achieving dynamic control over catalytic systems is the implementation of electric fields. Catalyst control using external electric fields has been studied extensively[14,15], and linear scaling relations between first-principles obtained adsorption energies at different external electric fields were observed in a recent study by Shetty et al.[16]. However, the magnitude of the homogeneous electric field required for substantial change in adsorbate binding energy (~1 V/Å) is hard to achieve experimentally in thermocatalytic reactors.

A more accessible method to electronically modulate catalyst active sites is the use of polarization of support materials. Oxides with high dielectric constants are commonly used to induce large charge concentrations in layers deposited on them[17]. As an application of this idea in catalysis, Wang et al. showed that $MoS_2$ deposited on the dielectric $SiO_2$ accelerated the rate of hydrogen evolution reaction (HER) by four-fold through an application of 100 V gate voltage in a catalytic field effect transistor[18]. More recently, a catalytic condenser stack device consisting of dielectric $HfO_2$ separating conductive electrode layer and catalyst loaded on graphene successfully demonstrated perturbation of the binding energy of adsorbed reactants for ultrathin film $Al_2O_3$ and Pt nanocluster catalysts[19,20]. In both of these devices, electronic structure perturbation at the active site increases with a higher dielectric constant. Materials with even higher dielectric constants can further alter the electronic structure of the active metal and accelerate the reaction rates.

Ferroelectrics are materials that exhibit spontaneous macroscopic electric polarization (electric dipole moment per unit volume) below a ferroelectric transition temperature[21]. Unlike dielectrics, in which the response to electric fields can be quantified by a constant permittivity, ferroelectrics have a richer electric field response[22]. When used in a capacitor-like device, ferroelectric materials lead to charge accumulation even in the absence of an externally applied potential difference[23]. The effect of the electric field on a ferroelectric material is to switch the polarization direction, and hence the sign of the charge accumulated in the electrodes. This nonlinear behavior and the large ferroelectric polarization-induced charge densities compared to those that can be achieved by dielectrics, make ferroelectric materials ideal candidates for catalyst surface control with low energy input.

The use of ferroelectrics to modify surface behavior and enhance catalytic activity has been studied extensively, both experimentally[24,25] and through first-principles simulations[26,27]. More recently, cyclic switching of support polarization as a means to surpass the limitations

of the Sabatier principle was explored for various systems[28], such as nitrogen oxide decomposition reaction on $CrO_2$-deposited $PbTiO_3$[29] and water splitting on bare $PbTiO_3$[30]. Also, the possibility of polarization breaking the scaling relations was discussed. However, previous studies in the literature relied on fixing at least part of the positions of ions, which is different from the structures optimized under a fixed-polarization constraint at a given polarization. To the best of our knowledge, there is no study that considers these optimized intermediate polarization structures to map out the reaction pathways' energies continuously as a function of polarization, unlocking potential for detailed analysis of polarization-scaling relations.

In a ferroelectric catalytic condenser (or capacitor) architecture, a ferroelectric insulating layer is incorporated in a capacitor stack architecture, where the polarization of the ferroelectric promotes electronic structure changes in the separated electrode and catalyst film layers. In this work, we computationally evaluate a Pt(100) thin film catalyst/electrode with $PbTiO_3$ insulator support to understand the quantitative relations between support polarization and properties overlaying Pt catalyst—changes to its geometry, electronic structure, the surface adsorbate binding, and reaction energies. We observe LSRs for adsorption energies for the same binding site but breaking of the TSS relations with a positive slope, where its origin is identified from the interface interaction between the support and the catalyst.

## Results and discussion
### Polarized structures
Figure 1b, c shows a conceptual difficulty in working with the polarization of crystalline solids[31]. While the definition of dipole moment for molecules is well-defined, that of solids is ambiguous depending on the choice of a unit cell. The modern theory of polarization answers this puzzle by showing that the polarization change compared to the non-polar state, rather than the polarization itself, is the more relevant physical property of a solid[32,33]. The ground state of crystalline materials without an applied electric field can be calculated using standard first-principles methods. At this state, ferroelectric materials are polar, whereas dielectric materials are non-polar. The polarization of the insulators will change if an electric field is applied, depending on the material's dielectric susceptibility. Over the past two decades, there have been rapid advances in methods of calculating the fixed-field responses of insulators in the literature, along with the advent of the modern theory of polarization[34–37].

However, while there are formally rigorous ways to calculate the dielectric response of metal/insulator heterostructures under applied constant electron fields from first-principles, these methods are often computationally demanding enough to be impractical for large

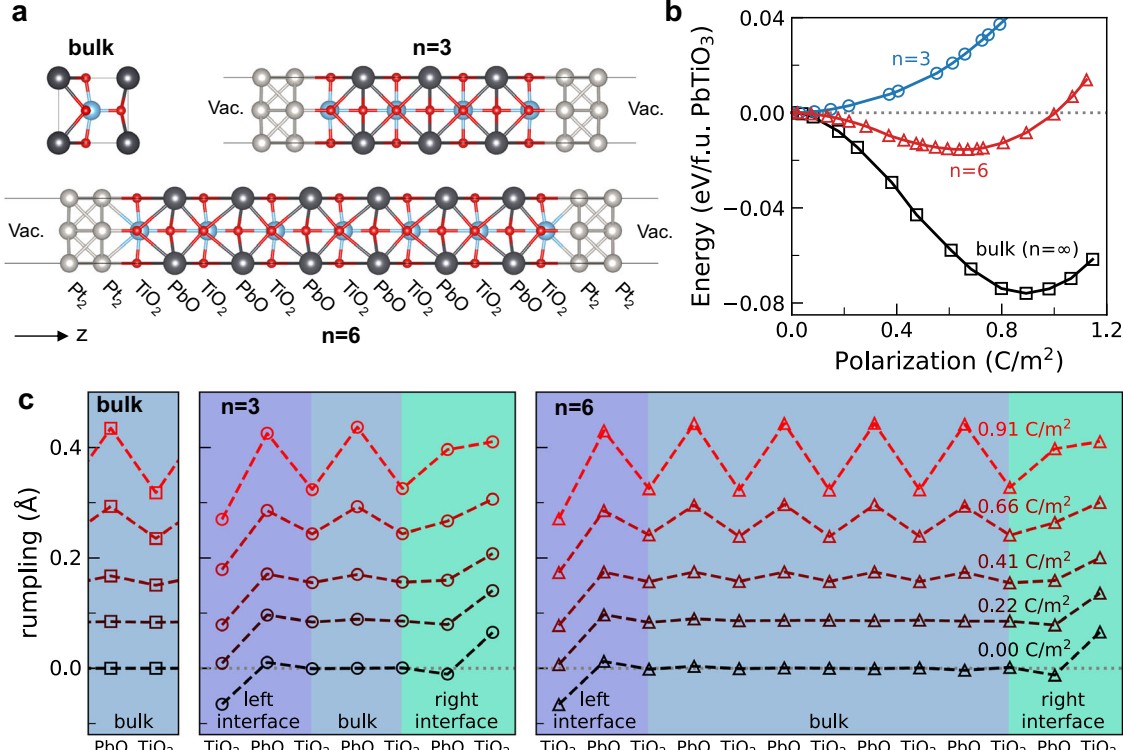

**Fig. 2 | Structure and energy variation with support polarization. a** Unit cell of bulk PbTiO$_3$ ($n = \infty$, top left), $(Pt_2)_2/TiO_2(PbOTiO_2)_3/(Pt_2)_2$ ($n = 3$, top right) and $(Pt_2)_2/TiO_2(PbOTiO_2)_6/(Pt_2)_2$ ($n = 6$, bottom) at their spontaneous polarized state. Atoms are represented as spheres of different colors, Pt: gray, Pb: black, Ti: blue, O: red. Vacuum space is represented as Vac. **b** Energy per formula unit PbTiO$_3$ at different polarization of bulk PbTiO$_3$, $n = 6$ and $n = 3$ structures. **c** Rumpling (difference in out-of-plane coordinate of anions and cations on a layer) of each layer for bulk PbTiO$_3$ (left), $n = 3$ (center) and $n = 6$ (right) at select values of polarization. Regardless of spontaneous polarization and display of ferroelectricity, the geometrical features of bulk and interfacial PbTiO$_3$ are alike at the same value of polarization.

systems and high throughput studies needed for catalysis research. For example, methods proposed by Stengel et al. utilize hybrid Wannier functions with separate treatment of states based on their electron occupancy[38], or half-cell geometries with introduced layers of bound charges[39]. In order to perform calculations more efficiently on vacuum-separated metal/insulator/metal heterostructures, we instead utilize a simpler approach, i.e., the constrained-forces method introduced by Sai et al. and Fu and Bellaiche[40,41]. The constrained-forces method of finding insulator ground states at constant polarization is simple to implement, fast, and applicable to virtually any first-principles code which allows calculating Born effective charges (BECs) and Hellman-Feynman forces. It is based on finding the ground state structure where Hellmann-Feynman forces on the ions are proportional to their BECs:

$$F_{i,\alpha} = -e\sum_{\beta} Z^{*}_{i,\beta\alpha}\mathcal{E}_{\beta} \qquad (1)$$

Here, **F** is the Hellman-Feynman force, $e$ is the elementary charge, **Z** is the BEC tensor, and $\mathcal{E}$ is the electric field. $i$ is the atomic index, $\alpha$ and $\beta$ are indices for Cartesian directions. A more detailed discussion of the constrained-forces method is presented in Supplementary Note 1.

The constrained-forces method offers several advantages, but there are some limitations and approximations to the method. One such assumption is that the BECs are considered constant irrespective of position and polarization (displacement field). Another shortcoming is that it does not involve differences in the Fermi levels of the two electrodes, as it would if external bias potential was used in experiments to induce polarization and field in the insulator. Despite these limitations, the effects of these approximations are negligible for

small fields, making the constrained-forces method a practical way to study the properties of systems under fixed polarization conditions.

In order to show that this method correctly captures the suppression of ferroelectricity in thin films, we performed constrained-forces calculations on three different systems shown in Fig. 2a: Bulk PbTiO$_3$, which is a single unit cell under periodic boundary conditions ($n = \infty$); a slab that consists of a heterostructure with 3 unit cells of PbTiO$_3$ with TiO$_2$ termination on both sides and 2 layers of Pt in the electrodes, $(Pt_2)_2/TiO_2(PbOTiO_2)_3/(Pt_2)_2$ ($n = 3$); and a similar slab with a thicker perovskite block $(Pt_2)_2/TiO_2(PbOTiO_2)_6/(Pt_2)_2$ ($n = 6$). PbTiO$_3$ is ferroelectric in bulk, and we calculate its spontaneous polarization as 0.89 C/m$^2$, which agrees with previously reported values of 0.88 C/m$^2$ from first-principles[42], and an experimental value of 0.75 C/m$^2$ at 295 K[43].

In Fig. 2b, we show the energy per formula unit PbTiO$_3$ as a function of polarization, relative to zero-field non-polar structure with lattice parameter $a$ fixed to the bulk value. The results show a significant effect of the depolarization field (and possible coherence volume effects[44,45]) for the thin-film ferroelectrics. As the ferroelectric thickness decreases, the non-ferroelectric interface capacitance dominates, which increases the inverse capacitance and decreases spontaneous polarization[46]. The critical thickness for the slab geometry is 6.5 unit cells ($n = 6$), and spontaneous polarization is suppressed below this thickness. As a result, the energy of the $n = 3$ slab increases with increasing polarization. The spontaneous polarization of the $n = 6$ sample is also reduced to 0.64 C/m$^2$ from the bulk value. However, it still displays the characteristic double-well energy-polarization curve, instead of a single-well centered at zero polarization. In passing, we note that the ferroelectric double-well energy landscape leads to a finite energy cost of flipping the polarization, which may be an

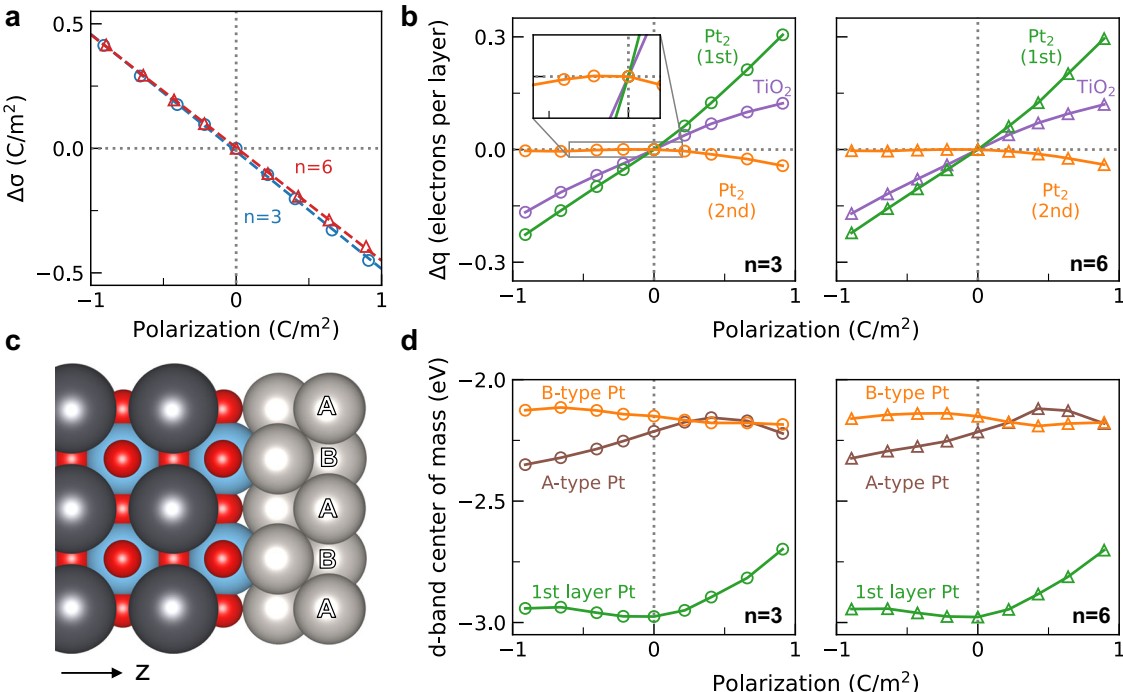

**Fig. 3 | Charge and band structure response to support polarization. a** Surface charge density difference of $n = 3$ and $n = 6$ structures at different polarization. Surface charge density scales linearly with polarization. **b** Bader charge difference per layer of interface $TiO_2$, interface 1st layer $Pt_2$ and 2nd layer $Pt_2$ for $n = 3$ and $n = 6$. **c** Difference between A-type and B-type Pt atoms on 2nd layer $Pt_2$. Atoms are represented as spheres of different colors, Pt: gray, Pb: black, Ti: blue, O: red. **d** $d$-band center of mass of different types of Pt atoms. Both charges and $d$-band center are a function of local geometry and polarization alone, regardless of ferroelectric properties, energy, or spontaneous polarization of the insulator.

important factor in practical applications. A rough estimate of this energy scale for bulk-like $PbTiO_3$ is ~0.1 eV per formula unit. Reducing the switching energy and coercive field (for example by strain engineering) is desirable not only for catalytic but also for other applications of ferroelectrics, and is an area of active research[47–49].

An interesting observation is that despite the differences in the energy vs. polarization curves, the structural characteristics of the slabs remain similar for equal polarizations. In particular, comparing the layer rumplings of $PbTiO_3$, except for the two to three layers close to the interface, there is no difference in the structure to the bulk ($n = \infty$). Also, between the two slabs with a thickness of three and six unit cells, the distortion pattern of the structure near the metal interface is almost identical at equal polarization (Fig. 2c). Considering that the approximations made with the constrained-forces method are not applied to zero-field calculations (spontaneous polarization for $n = \infty$ and $n = 6$), this shows that the constraint-forces method results in a reasonable and fast calculation for heterogeneous metal-insulator systems at fixed polarization.

**Electronic structure changes**

The structural resemblance between 6.5 unit cells $PbTiO_3$ ferroelectric slab and 3.5 unit cells $PbTiO_3$ dielectric slab at corresponding polarization justifies considering electronic properties of the catalyst to be solely a function of local geometry, and polarization alone despite the depolarization fields. To further verify this, we conducted an analysis of the electronic structure of the Pt catalyst overlayer, with results shown in Fig. 3. Among the properties of metal catalysts that are known to affect adsorption, we show Bader charge[50,51] (see Supplementary Fig. 4 for DDEC6 charge) and the so-called $d$-band center of mass (weighted center of the occupied $d$-band densities of states)[52,53].

The changes of charges of each ion are shown in Fig. 3a, b. The sum of the charges accumulated in the electrode and the interface depends linearly on polarization, both for 6.5 unit cells and 3.5 unit cells lead titanate structures (Fig. 3a) with a slope of −0.47. The free charge surface density induced should scale one-to-one with the displacement field, which depends on the polarization. However, there is no well-defined method of separating free charges from bound charges in first-principles calculations, and our estimate of induced charges contains the effects of bound charges.

While the net change of charge is linear and monotonic, the charge resolved by each layer exhibits a different trend. Figure 3b shows layer-resolved differences in charge for 3 layers, the interface $TiO_2$ layer, the first layer of $Pt_2$ (interface), and the second layer of $Pt_2$ (surface). Interface $TiO_2$ has conductive properties from metal-induced gap states (MIGS). The layers at the interface, the $TiO_2$ layer, and the first layer of $Pt_2$ depict a monotonic increase in the number of electrons with increasing polarization. However, the second layer of $Pt_2$ shows non-monotonic behavior with a much smaller magnitude; even toward the opposite direction of charge accumulation of interface. This is against what can be simply expected from a simple picture of "doping" the active sites with charges for ultra-thin film catalysts, where the charge induced on each metal atom has the same sign. This is due to the electric field screening effect of metallic platinum, where even a single layer of atoms almost completely screens the electric field induced by the support. In passing, we note that there is almost no difference in the trends of the Bader charges between 6.5 unit cells and 3.5 unit cells $PbTiO_3$ structures, which means we can use constrained-forces method for smaller systems with thinner insulator to effectively simulate larger systems at lower computational cost.

The changes in the densities of states and the $d$-band center of mass[54] for each chemically unique Pt atom are shown in Fig. 3d. Compared to Bader charges, the $d$-band center of mass exhibits a more complicated trend against polarization. The changes are non-monotonic and distinctive for each Pt, even within the same layer. (The two Pt atoms on the second layer are not chemically equivalent to each other: there are two different environments, depending on

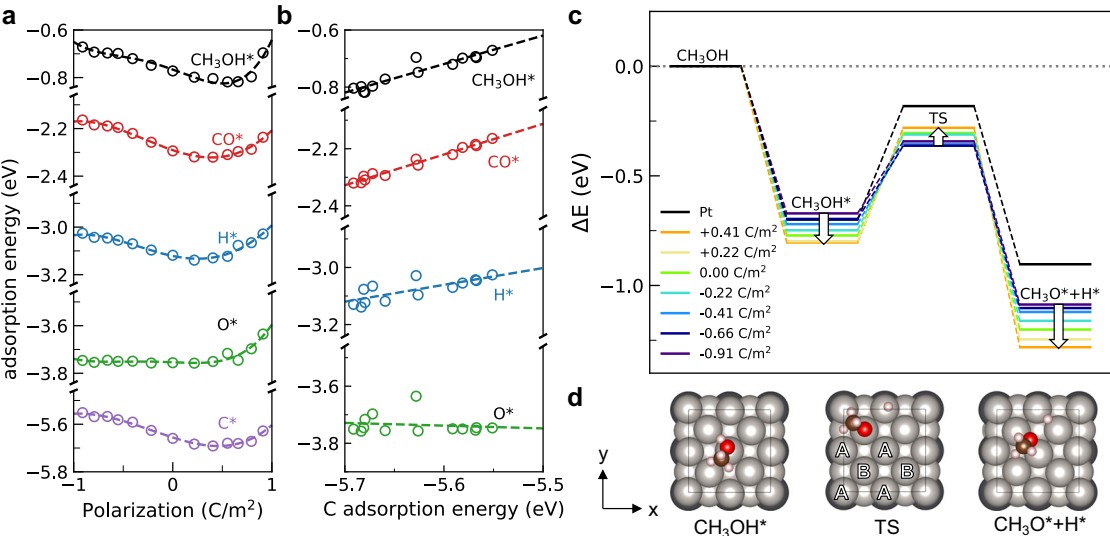

**Fig. 4 | Thermodynamic and kinetic catalyst control with support polarization.** **a** Changes of adsorption energies on top sites of A-type Pt for CO*, CH₃OH*, H*, O*, and C* with varying polarization. **b** The adsorption energies scale linearly with the adsorption of carbon atoms. **c** Reaction coordinate of O-H bond dissociation of methanol at Pt(100) surface, on various polarized PbTiO₃. As polarization of support increases from $-0.66$ C/m² to 0.41 C/m², the energy of IS and FS decreases, whereas the energy of TS increases. **d** Positions of dissociating methanol on the Pt(100) supported on PbTiO₃ surface at the initial state, transition state, and final state of O-H bond dissociation.

whether they are aligned with a Pb or a Ti cation in the PbTiO₃ substrate. We refer to them as the A-type and B-type Pt.) Most importantly, the second layer $d$-band centers of mass show more significant changes compared to changes in the Bader charges, which may explain the changes in the reaction energy barriers discussed below. As before, while the $d$-band center for each Pt behaves differently, the overall trends are similar for 6.5 unit cell and 3.5 unit cell lead titanate structures within the range of error.

**Adsorption and reaction control**

The results in Figs. 2 and 3 demonstrate that the properties of oxide-supported catalyst surfaces can be changed by the polarization of the supporting insulator and that this change is independent of insulator thickness. This agrees with the observations of ref. 26, where only two values of polarization for each thickness were considered. This suggests that the electronic structure of the catalyst surface is only affected by the polarization and the local geometry of the few interfacial layers; but not by the structure over three to four layers deep within the oxide. The interaction of adsorbate with the surface can also be expressed as a function of polarization, as long as the interface and the surface stay the same. Minor changes occur with oxide thickness differences, due to the work function changes after the adsorption; these small changes will be ignored in this context (see Supplementary Fig. 6).

In Fig. 4, we show the adsorption energies of select adsorbates C*, O*, H*, CO*, and CH₃OH* on the top site of A-type Pt for varying degrees of polarization (see Supplementary Fig. 7 for different adsorption sites). A more negative value of adsorption energy represents a stronger interaction. While the specific shape of the curves in Fig. 4a differ, the adsorption energies commonly increase along with the increase in polarization from $-1.0$ C/m² to 0.5 C/m² and then decrease again up to 1.0 C/m². Both CO* and CH₃OH* adsorption energy curves against polarization are similar to that of C*, which also agree with the $d$-band center of mass trend of A-type Pt (see Supplementary Fig. 8). The adsorption energies scale with each other, as shown in Fig. 4b. We don't, however, observe a correlation between the adsorption energies and Bader charges, or specific orbitals centers[26,55].

The non-monotonic change in adsorption energies as a function of polarization entails that the largest difference in the surface

properties between opposite polarization states is not realized when the polarization is the largest. Instead, there is an intermediate value of polarization, for example, -0.5 C/m² for CH₃OH*, where the largest modulation of adsorption energy is obtained under polarization switching. This makes the design of ferroelectric materials where the polarization magnitude is tuned to the desired value through strain, doping, or layering an important first step to realizing superior catalytic capacitor devices.

The transition states of O-H bond breaking in methanol decomposition were found to exhibit unique scaling relations on the platinum/lead titanate/platinum system perturbed by support polarization. Figure 4c shows the reaction coordinate of the minimum energy pathways (MEPs) for the breaking of the O-H bond of methanol between $-0.91$ C/m² and 0.41 C/m². Rate-determining steps for alternate pathways of methanol decomposition were not considered in this study in order to focus on demonstrating the effect of support polarization. On higher positive polarization, an MEP under the convergence criterion was not found, indicating a large modification of the reaction pathway. This regime can be avoided by tuning the spontaneous polarization strength of the oxide and hence does not necessarily constitute a drawback. On bare Pt slab, O-H dissociation of methanol has an initial state (IS) energy of $-0.70$ eV, transition state (TS) energy of $-0.18$ eV, and final state (FS) energy of $-0.90$ eV. PbTiO₃-supported two layers of Pt(100) at zero polarization change the IS energy to $-0.77$ eV, TS energy to $-0.31$ eV, and FS energy to $-1.20$ eV. Within the support polarization range of 0.41 and $-0.66$ C/m², TS energies and IS, and FS energies scale with a negative slope. Initial state and final state energy decrease with decreasing polarization from ~0.5 C/m² and transition state energy increases. As support polarization decreases further, we observe that the TS energy increases again but the activation energy is still lowered. The overall effect of support polarization lowers activation energy significantly from 0.52 eV (+0.41 C/m²) to as low as 0.33 eV ($-0.91$ C/m²).

The changes in absorption and transition state energies reveal two important points. First, the magnitudes of changes at the catalyst surface are consistent with the requirements of programmable catalysts for dynamic rate control[10]. Variation in the extent of PbTiO₃ polarization can be achieved via changing the material properties by

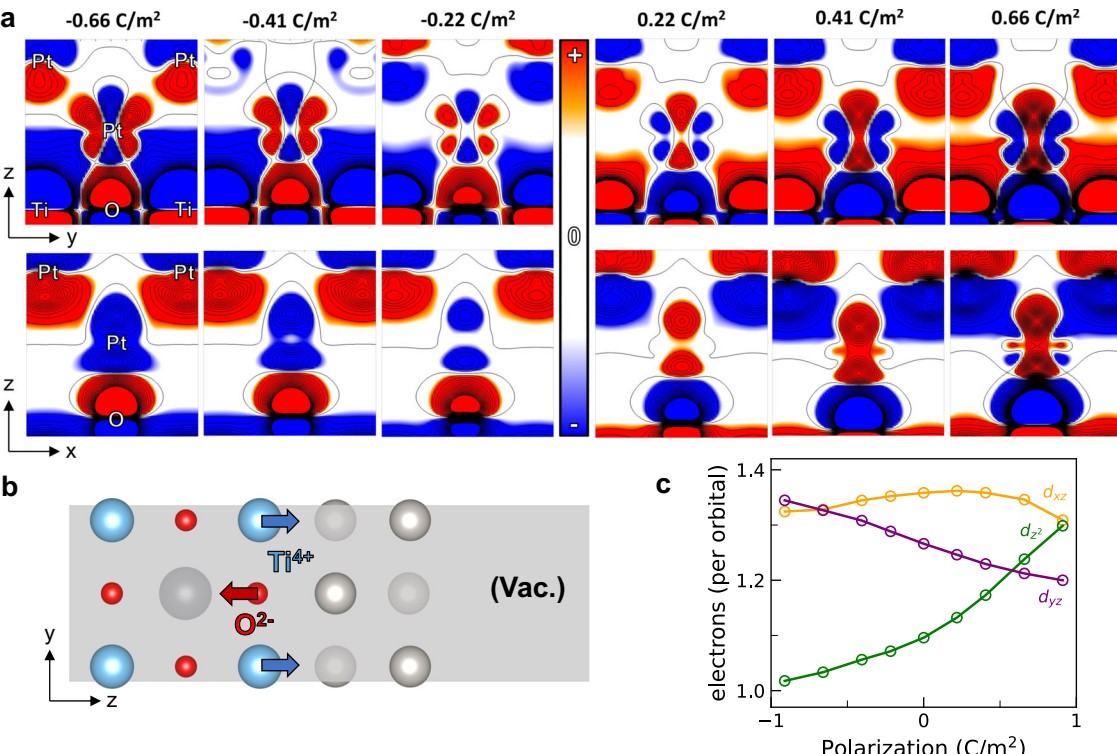

**Fig. 5 | Orbital-specific charge density response to support polarization. a** Real-space charge density difference plots, compared to non-polar structure. The top row represents the $yz$ plane, the plane direction toward B-type Pt. The bottom row represents the $xz$ plane, the plane direction toward A-type Pt. As polarization increases, the $d_{z^2}$ orbital electron population increases. The presence of Ti ions in the $yz$ plane induces drastic changes of $d_{yz}$ orbital electron population, while no visible changes are made to $d_{xz}$ electrons. **b** Displacement of interface ions with increasing polarization. Ti cation moves toward, and O anion moves away from Pt. **c** Integrated d-orbital electron population from PDOS of 1st layer Pt atom at different polarization of support PbTiO₃.

strain, doping, layering, etc. as has been commonly done in perovskite-related oxides, and the polarization can be switched via an external potential bias applied to Pt layers acting as electrodes[56–59], thereby configuring the metal catalyst electronic properties via an external input with time (i.e., a program). Dynamic controlled modulation of catalyst electronic state via external bias has been demonstrated to achieve resonance conditions that exist beyond conventional static catalytic rates[11,13,60]; this resonance mechanism occurs only for catalysts that can be externally controlled to modulate the catalytic surface energy with time[61,62]. For the considered methanol decomposition system, variation of device (Pt/PbTiO₃/Pt) polarization yields a significant change in the binding energy of key reaction intermediates, indicating that metal-ferroelectric devices provide the capability for transient modulation to control the reaction surface chemistry via input voltage programs.

The second point is that the positive TSS relations are broken. The change in support polarization provides an additional degree of freedom in the scaling relations. We find the breaking of the positive TSS relations to be mainly driven by the inherent geometry of the oxide, affecting the Pt overlayer uniquely depending on the active site location. In the previous section, it was observed that there are two different types of Pt atoms on the second platinum layer, the A-type and B-type. As the $d$-band center dependence on polarization for each type of Pt is different, so are the adsorption energies for the A and B sites. Since the changes in adsorption energy with polarization are different for each site, it is possible for reactions involving multiple adsorption sites to have negative scaling relations. For example, one of the MEPs for the methanol O-H bond dissociation reaction involves multiple adsorption sites; it begins with CH₃OH* adsorbed at the top site of the A-type Pt atom. At the transition state, the O-H bond cleaves, and CH₃O* is bonded to the B-site Pt atom while H* is to the top site of

A-type Pt. Finally, at the final state, both the CH₃O* and H* rest at the bridge sites (Fig. 4d).

We now investigate the origin of behavioral differences of Pt atoms from the same layer in more detail, using real-space charge density. Figure 5a shows how the charge density changes as a function of polarization, relative to the non-polar structure. Since the change in polarization involves changes in ionic positions as well, the plots are drawn by keeping a first-layer Pt atom in the center. Charge density changes visible for atoms other than the first-layer Pt mainly involve displacement of core electrons and convey less information. The upper row shows the plots for the $yz$ plane, which includes second-layer A-type Pt atoms. The bottom row shows the plots for the $xz$ plane and includes second-layer B-type Pt atoms. The main difference is that there are Ti cations in the $yz$ plane of first-layer Pt, and not in the $zx$ plane. While increasing polarization increases the population of $d_{z^2}$ orbitals of first-layer Pt, it does not affect $d_{yz}$ and $d_{xz}$ equally. There is a clear indication of a decreasing population of $d_{yz}$ orbitals electrons, from decreasing distance of Pt-Ti as polarization increases. On the contrary, there is no clear evidence of $d_{xz}$ orbital electrons being affected by polarization. This difference drastically affects A-type and B-type sites, as upper $d_{yz}$ lobes point directly toward the A-type platinum. For example, it can be observed that as polarization increases from zero, the A-type Pt atom moves upwards, and the B-type Pt moves downwards relative to the first-layer Pt from the reversed color scheme near the second-layer Pt atoms.

This discussion suggests that the metal-insulator interface is possibly the most important factor determining electronic structure at the exposed catalytic surface, at least in the ultra-thin catalyst limit. As polarization increases, the rumpling of the interfacial oxide layer changes, which moves either Ti cations or O anions closer or further away from the interface Pt. (Fig. 5b) The changes at interface Pt can be

approximated as a crystal field response, generated from the displacement of interface ions with polarization changes. Integrated projected density of states (PDOS) further confirms this point, as shown in Fig. 5c. The number of electrons on $d_{z^2}$, $d_{yz}$, and $d_{xz}$ is plotted against polarization. There is a clear trend of an increasing number of $d_{z^2}$ electrons and a decreasing number of $d_{yz}$ electrons, consistent with real-space charge density plots. The $d_{xz}$ exhibits non-monotonic changes in smaller magnitude compared to $d_{yz}$ and $d_{z^2}$ electrons.

Dynamic programmable catalysis is an emerging field that offers a promising approach to overcoming the limitations of the Sabatier volcano by periodically modulating the properties of a catalyst. This revitalizes the interest in ferroelectric capacitors with ultrathin metal catalysts, as a candidate for this type of catalysis. In this study, we provided a detailed discussion of the thin-film Pt catalyst on ferroelectric PbTiO₃ in the static limit of the crystal, which corresponds to frequencies lower than the typical lattice vibration frequencies in the terahertz range. We first proposed a way of simulating the capacitor at fixed support polarization efficiently using the constrained-forces method. We studied geometry, atomic charges, and the electronic structure of the catalyst for different values of support polarization. We revealed that the properties of the catalyst are a function of support polarization, which is not limited to the spontaneous polarization of the device or bulk material. We then investigated the origin of changes in adsorbate-surface interaction within the context of $d$-band theory. Finally, we observed that methanol decomposition deviates from positive transition state scaling. This result, combined with emerging studies of dynamic microkinetic modeling[63] will provide a mechanism for faster, dynamic surface catalysis. We also identified an intermediate value of polarization as the optimal point for the highest modulation, emphasizing the importance of the computational study of the metal-oxide interface in detail. Our work lays the foundation for future optimization of the catalytic capacitor system, including the interface and surface, to achieve the highest possible turnover.

## Methods

Periodic density functional theory (DFT) calculations were performed with a customized version of Vienna ab initio simulation package (VASP) 5.4.4[64] modified for constrained-forces calculations. Spin-polarized calculations were performed for sample systems with no different outcomes. A generalized gradient approximation (GGA) exchange-correlation functional with non-local van-der-Waals (vdW) correction, optB86b-vdW functional by Klimeš et al. was used[65]. The functional was chosen as a compromise between the accurate prediction of the lattice parameters and the adsorption energies[66–68]. The projector augmented wave (PAW) method to describe atom cores and the plane wave basis set was expanded to a kinetic energy maximum of 520 eV for Kohn-Sham orbitals. A Gaussian smearing profile of 0.05 eV was imposed on electrons at the Fermi level, and energies were extrapolated to zero smearing. The climbing image nudged elastic band method[69] was used to estimate the MEPs and transition states.

### Bulk structure calculations

Tetragonal lead titanate (PbTiO₃) with space group: P4mm was used as a model ferroelectric material for computations. For bulk structure calculations, full optimization of the polarized structure was converged below $10^{-9}$ eV of electronic energy changes and $10^{-4}$ eV/Å of maximum ionic forces. A $9 \times 9 \times 8$ Gamma-centered **k**-point mesh was used to sample the Brillouin zone. The obtained lattice constants $a$ and $c$ were 3.906 Å and 4.157 Å, which agreed with experimental results of 3.905 Å and 4.151 Å[70]. Born effective charges (BEC) were obtained using linear response theory on bulk PbTiO₃ without polarization (space group: P4/mmm), with lattice constant $a$ fixed to 3.906 Å. The polarization of structures and each oxide layer were estimated using BECs and local atomic displacements. To account for the net charge for each oxide layer not being zero, the dipole moment of each layer was

calculated from the charge center of each layer. Resulting layer dipole moment $\mu_l$ can be expressed as:

$$\mu_l = -\frac{2Z_n^* Z_p^*}{Z_p^* - Z_n^*} d \tag{2}$$

where $Z_n^*$ is the sum of negative BECs, $Z_p^*$ is the sum of positive BECs, and $d$ is the rumpling, the positional difference of metal ion and oxygen ion toward the direction of the dipole moment. The sum of the dipole moments was divided by the volume of the oxide to obtain effective polarization. Finally, the 1st order correction term $(1 + \chi_\infty / \chi_{\mathrm{ion}})$ was multiplied to effective polarization to estimate the value of polarization, $P$. The correction term value of 0.93 was used for PbTiO₃[39].

### Slab heterostructure and band structure calculations

Slab calculations were performed with two or three layers of Pt selected as the active catalyst layers, and three-to-six unit cells of TiO₂-terminated PbTiO₃ between the Pt layers[26]. A unit cell of slab structure can be expressed as $(\mathrm{Pt}_2)_m/\mathrm{TiO}_2(\mathrm{PbOTiO}_2)_n/(\mathrm{Pt}_2)_m$ with $m$ equal to two or three and $n$ between three and six. Only single-domain, single-crystal structures were considered for this study. Slabs are symmetric at zero polarization to remove any unwanted external Volta field arising from work function differences. The lattice parameter, $a$, was fixed to 3.906 Å. Pt lattice parameter at ground state is 3.948 Å, and biaxial compressive strain of 1.06% is applied to Pt. Geometries were considered optimized when electronic energy changes were below $10^{-6}$ eV and the maximum ionic forces (difference from target forces) were below 0.005 eV/Å. Γ-centered **k**-point mesh of $11 \times 11 \times 1$ was used. Over 15 Å of vacuum was used to prevent interaction in the z-direction, and dipole correction in the middle of the vacuum was used to eliminate the artificial electric field and unphysical dipole-dipole interactions among the copies in the z-direction for slab calculations[14]. The PAW approach we use imposes periodic boundary conditions in all three directions. To calculate the density of states (DOS), Bader[51] and DDEC6[71] charge of atoms, FFT grid number of four times the kinetic energy cutoff in each lattice direction was used (~28,570 grid points/Å³) with 0.10 eV Gaussian Smearing and $21 \times 21 \times 1$ **k**-points sampling. Wigner-Seitz radius used for PDOS is 1.46, 0.95, 0.95, 1.11 Å for Pb, Ti, O, and Pt based on the smallest Bader volumes of each element. The number of Kohn-Sham orbitals were kept greater than 0.9 times the number of electrons in the system. $d$-band center of mass, $E_{d\text{-COM}}$ is calculated as weighted average energy level of $d$-band occupied states up to Fermi level, $E_F$, relative to Fermi level:

$$E_{d-\mathrm{COM}} = \frac{\int_{-\infty}^{E_F} \rho(E) E \, dE}{\int_{-\infty}^{E_F} \rho(E) \, dE} - E_F \tag{3}$$

where $\rho(E)$ is electronic density of states, and $E$ is energy.

### Adsorption energy and transition state calculations

Adsorption energies were calculated on a surface of $(2\sqrt{2} \times 2\sqrt{2})$ Pt (100) surface (1/8 monolayer coverage) supported on polarized polarized PbTiO₃. Platinum layers at the surface of adsorption and the top two layers of PbTiO₃ (one TiO₂ layer and one PbO layer) were allowed to relax, and the remaining structure was fixed. Structures were optimized until the norms of all ionic forces were below 0.015 eV/Å with Gamma-centered **k**-point mesh of $9 \times 9 \times 1$. Adsorption energies $E_{\mathrm{ads}}$ were obtained by subtracting relaxed slab energy $E_{\mathrm{slab}}$ and adsorbate energy $E_{\mathrm{adsorbate}}$ from adsorbed complex $E_{\mathrm{complex}}$:

$$E_{\mathrm{ads}} = E_{\mathrm{complex}} - E_{\mathrm{slab}} - E_{\mathrm{adsorbate}} \tag{4}$$

(100)-hex reconstruction of the surface[72] or octahedral rotation of perovskite was suppressed. Transition state energies were obtained

using the climbing image nudged elastic band (CI-NEB) method from Transition State Tools for VASP (VTST) 3.2, with three to five intermediate images excluding initial and final states[69]. The images were considered converged once the absolute maximum of all forces perpendicular to the band were below 0.03 eV/Å.

Two different energy pathways for O-H bond dissociation of methanol on Pt (100) slab that satisfies the convergence criterion have been found, using 4 intermediate images. For the energy pathway with lower activation energy, the CI-NEB method with three to five intermediate images converged with the same activation energy, within 0.01 eV. The minimum energy pathway on the Pt (100) slab was used as the initial guess for CI-NEB on $PbTiO_3$ supported Pt (100), using 3 intermediate images.

### Crystal structure visualizations
VESTA software[73] was used for building and visualizing crystal structures.

### Data availability
Source data are provided with this paper.

### Code availability
Code is available at https://doi.org/10.5281/zenodo.10045191.

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

## Acknowledgements

S.J., C.P., P.B., P.J.D and T.B. are supported by CASALE SA. P.J.D. acknowledges support from the Center for Programmable Energy Catalysis, a U.S. Department of Energy Energy Frontier Research Center (DE-SC0023464). T.B. and S.J. acknowledge partial support from the Office of Naval Research Grant (N00014-20-1-2361), and computational resources from the Minnesota Supercomputing Institute (MSI) at the University of Minnesota, Anvil at the Purdue University and ACCESS for providing resources that contributed to this research.

## Author contributions

S.J., C.P., P.B., P.J.D. and T.B contributed to study conception. S.J. performed calculations, collected data and analyzed the results under supervision of T.B., S.J., P.J.D. and T.B. contributed to preparation of manuscript from input from all authors.

## Competing interests

The authors declare no competing interests.
