## [Peer Review File · Nature Communications]

Reviewers' Comments:

Reviewer #1:

Remarks to the Author:

The authors present a DFT study that seeks to understand the effect of support polarization on the properties of a supported catalyst thin film, using ferroelectric PbTiO₃ and Pt thin film (bilayer) as a model system. The central claim of the study is that support polarization — in particular, the asymmetric changes in the environment imposed by the underlying structure of the support on the catalyst thin film — can help break transition state scaling relations. The underlying ideas are certainly interesting and the results are potentially significant, modulo some technical details that need to be addressed more thoroughly/precisely before the overall validity of the claims can be assessed.

Comments:

1) The model system consists of a $2\sqrt{2} \times 2\sqrt{2}$ Pt(100) bilayer on PbTiO₃. First, were the slab and thin film constructed to be commensurate? If not, what is the strain imposed on Pt by the PbTiO₃ substrate and to what degree does this affect the energetics? Second, the surface supercell looks to be rather small (as seen in Fig. 4d). How sensitive are the results to finite-size effects? The adsorption energies might not yet be sufficiently converged with system size especially when multiple adsorption sites are involved in the dissociation steps. This needs to be addressed carefully.

2) A central claim of the work is that while the adsorption energies of the initial and final states decrease with decreasing polarization, the transition state energies increase. The activation barrier changes by about 0.2 eV across the range of polarizations studied here. However, I see from the SI that the transition state calculations (CINEB) use a rather high force tolerance of 0.045 eV/Å (unlike normal relaxations at 0.02 eV/Å). With such high residual forces, coupled with the previous question about finite size effects, it is hard to conclude whether this significant change (0.2 eV) is a computational artifact or real.

3) For the magnitude of surface electric field implied by the support polarization, what is the comparable response of a pure Pt(100) slab? This "control experiment", as it were, is missing from the studies and/or the discussion.

4) The authors use a constrained force method to determine the ground state of the ferroelectric slab under a given external field. I appreciate that this approach is (a) fast and (b) allows one to control polarization rather than electric field in the calculation. Yet, this approach does miss effects of polarization of electronic wavefunctions and, while this might have little effect on piezoelectricity where ionic dipoles dominate (as pointed out by Fu and Bellaiche), such effects might not be negligible when considering the response of the Pt layers (a metal, hence, highly polarizable). Given that it is fairly straightforward to do a slab calculation with an electric field (sawtooth potential along z , which is essentially what Eqs. 11-13 in the SI imply), some direct quantification of how good the approximations of the constrained force method — at non-zero field — are relative to a conventional sawtooth potential approach would lend more credence to the results.

Parenthetically, the constrained-force approach is not standard in VASP, as far as I'm aware (but I could be mistaken). At any rate, if the authors used their own modified version of the code, this should be stated and some text provided in the SI on how this is implemented.

Reviewer #2:

Remarks to the Author:

Authors are to be complimented on an exceptionally well written exposition on the potential influences of ferroelectric polarization on adsorption and reaction at on a metal/ferroelectric heterostructure. The Introduction does a very nice job of placing the work in the context of the field. I have only a few minor comments for consideration in a revision (in no particular order):

1. The inference from the calculations is that external polarization can induce different responses

of reactant and transition states (Figure 4d). It would be helpful in the Conclusions to draw a more direct line between this observation and the concept of a "programmable" catalyst. Could the effect be leveraged statically? Or dynamically? Both? Some discussion would help clarify what is currently a disconnect between the DFT observations and catalytic implications. Relevant to title as well.

2. Induced polarization comes with an energy cost. Can this be estimated, and what is its magnitude relative to a chemical energy?

3. Model assumes fixed epitaxy between Pt overlayer and support and (I believe) constant lateral directions. What is lateral strain on system, and is it sensitive to polarization? Asked another way, what is interface energy and how does it change with polarization? Is the epitaxy assumption reasonable?

4. Along same lines, is the proposed concept limited to a planar heterostructure geometry? Is it applicable to the supported metal nanoparticle motif more common in catalysis.

5. Figure 4 shows MEP is ~fixed vs polarization, yet comments on page 8 indicate that path changes at some polarization. Could authors comment further? As what appears to underlie the observations are differential influences on Pt atoms, one might well imagine changes in reaction pathway to be a significant and confounding effect working against the "programmable" catalyst.

6. Page 4, the $n=3$ and $n=6$ cases are inconsistent between the Figure and the narrative at the bottom of the page.

7. Unless I missed it, Figures 1b and 1c were not referred to in the text.

8. While motivation is presented as desire to increase TOF, selectivity and stability are often equally/greater importance in practice. Maybe tangential to the work here, but importance considerations in wrt the "programmable" concept.

Response to Referee 1

The authors present a DFT study that seeks to understand the effect of support polarization on the properties of a supported catalyst thin film, using ferroelectric PbTiO₃ and Pt thin film (bilayer) as a model system. The central claim of the study is that support polarization — in particular, the asymmetric changes in the environment imposed by the underlying structure of the support on the catalyst thin film — can help break transition state scaling relations. The underlying ideas are certainly interesting and the results are potentially significant, modulo some technical details that need to be addressed more thoroughly/precisely before the overall validity of the claims can be assessed.

Author Response: We thank the reviewer for their positive assessment on our work.

1) The model system consists of a $2\sqrt{2} \times 2\sqrt{2}$ Pt(100) bilayer on PbTiO₃. First, were the slab and thin film constructed to be commensurate? If not, what is the strain imposed on Pt by the PbTiO₃ substrate and to what degree does this affect the energetics? Second, the surface supercell looks to be rather small (as seen in Fig. 4d). How sensitive are the results to finite-size effects? The adsorption energies might not yet be sufficiently converged with system size especially when multiple adsorption sites are involved in the dissociation steps. This needs to be addressed carefully.

Author Response: The lattice parameter a used for model system is 3.906 Å, which corresponds to zero strain for tetragonal ground state of PbTiO₃. Lattice parameter for ground state Pt is 3.948 Å, which corresponds to about 1.0 % compressive strain. This information has been added to SI.

Strain is known to affect adsorption according to the d -band model (Mavrikakis et al. (1998) *Physical Review Letters*, 81(13), 2819.) and the eigenstress model (Khorshidi et al. (2018) *Nature Catalysis*, 1(4), 263-268.). However, as long as the strain remains constant and no significant changes occur in stress applied by adsorption, the effect of strain from both models should cancel when calculating the adsorption energy changes due to support polarization.

While there are not many reports of DFT studies of methanol adsorption on Pt(100), $2\sqrt{2} \times 2\sqrt{2}$ Pt(100) has larger surface area ($8a^2$) compared to 3×3 Pt(111) surface ($9\sqrt{2}a^2/2$) widely used to study methanol adsorption and decomposition.

We also add a calculation that shows methanol decomposition pathway and energetics on pure $2\sqrt{2} \times 2\sqrt{2}$ Pt(100) is converged within 0.02 eV compared to 3×3 Pt(100):

Cell size	Binding energy (eV)	Activation energy (eV)	Reaction energy (eV)
$2\sqrt{2} \times 2\sqrt{2}$	-0.681	0.499	-0.222
3×3	-0.697	0.493	-0.234

As can be seen from this table, while the quantitative values differ between the two different cell areas used, these changes are small (typically of a few percent), and don't change the conclusions drawn in the manuscript.

2) A central claim of the work is that while the adsorption energies of the initial and final states decrease with decreasing polarization, the transition state energies increase. The activation barrier changes by about 0.2 eV across the range of polarizations studied here. However, I see from the SI that the transition state calculations (CINEB) use a rather high force tolerance of 0.045 eV/Å (unlike normal relaxations at 0.02 eV/Å). With such high residual forces, coupled with the previous question about finite size effects, it is hard to conclude whether this significant change (0.2 eV) is

a computational artifact or real.

Author Response: In order to confirm that stricter tolerance factors don't make a difference and bring the quality of the numbers we report to the level that the referee requested, we have now reduced the tolerance of CI-NEB calculations from maximum force of 0.045 eV/Å to 0.030 eV/Å, and adsorption calculations from maximum force of 0.020 eV/Å to 0.015 eV/Å. The overall trend in both the adsorption and the transition state energies remained the same, except at the largest values of support polarization in the negative direction (-0.91 C/m^2) where the transition state energy increases slightly compared to previous values. This is likely a large enough polarization value that strong nonlinear effects emerge, as expected.

3) For the magnitude of surface electric field implied by the support polarization, what is the comparable response of a pure Pt(100) slab? This "control experiment", as it were, is missing from the studies and/or the discussion.

Author Response: We thank the referee for raising this point. However, we believe that a better "control experiment" for the system we study is Pt atoms placed on a non-polar oxide that does not create a modulated electric field on it. This corresponds to the zero polarization state of our system, and results regarding it is already present in the manuscript.

The effects of external electric field in plain Pt slabs, including Pt(100) slab have been studied in detail in the other literature, without any oxide [Shetty et al., (2020). *ACS Catalysis*, 10(21), 12867-12880]. However, we don't consider that study as a good control for the present work either since in the system we consider, there is no surface electric field (outside the slab). Our methods simulate internal electric field within the capacitor geometry, and electric field is only applied to the insulator between the electrodes, which corresponds to the application of a potential difference between the electrodes. The two approaches also differ in where free charge is induced. In the external electric field case it would be the surface where the reaction occurs, whereas in the internal electric field case it is at the metal/insulator interface. While external electric field may appear advantageous as it induces charge on the surface, it is hard to utilize it because it requires much larger electric potential to create electric field in free space. Utilizing internal electric field is not limited by the scale of the reactor and can take advantage of material properties such as large dielectric constant or ferroelectric response.

The only independent variables in our case is the polarization of the support and the thickness of the insulator. In demonstrating effects of one variable, the other was fixed as control variable. We have shown the effects of the thickness of the insulator by fixing the polarization and only varying the insulator thickness. We demonstrated that the insulator thickness, which controls the spontaneous polarization of the slab does not affect structure of both ions and electrons. We also showed effects of support polarization with fixed length of the insulator. In both demonstrations, we believe that the pure Pt(100) slab cannot be considered as a control experiment. Removing the support or using external electric field corresponds to a different experiment.

4) The authors use a constrained force method to determine the ground state of the ferroelectric slab under a given external field. I appreciate that this approach is (a) fast and (b) allows one to control polarization rather than electric field in the calculation. Yet, this approach does miss effects of polarization of electronic wavefunctions and, while this might have little effect on piezoelectricity where ionic dipoles dominate (as pointed out by Fu and Bellaiche), such effects might not be negligible when considering the response of the Pt layers (a metal, hence, highly polarizable). Given that it is fairly straightforward to do a slab calculation with an electric field (sawtooth potential along z, which is essentially what Eqs. 11-13 in the SI imply), some direct quantification

of how good the approximations of the constrained force method — at non-zero field — are relative to a conventional sawtooth potential approach would lend more credence to the results.

Author Response: Slab calculation with applied sawtooth electric field is easier to calculate, but it will not give desired results in our geometry of interest because it corresponds to external electric field where our system is enclosed by close-circuited metals. It will shield external electric field by reallocating free charges at the surfaces, which is not desired in our geometry where free charges are induced at the interface between the conductors and insulator. This is illustrated well in previous study from Stengel and Spaldin. (Stengel, M., & Spaldin, N. A. (2007) *Physical Review B*, 75(20), 205121. Fig. 2) Inducing internal electric field in the insulator in metal-insulator heterostructure for DFT calculations is much harder for this reason, with only few known methods that are not commercially available and hard to implement.

As the reviewer points out, the second-order electronic response of insulator wavefunctions is not captured using the constrained-forces method. However, Pt response to first-order ionic and electronic changes in the insulator is well captured through self-consistent calculation cycles of the wavefunctions. Also, the metal-insulator interface is already under high electric field from contributions such as electrostatic line-up term and interface capacitance. Compared to this value and first-order changes, contribution from second-order polarizable wave functions from small electric field is indeed negligible.

It is also worth pointing out that the electric field in our range of calculation is much smaller compared to polarization, with the largest magnitude of electric field used being $\epsilon_0\mathcal{E} = 0.0022 \text{ C/m}^2$ ($\mathcal{E} = 0.025 \text{ V/\AA}$). It practically makes polarization and displacement field at the insulator identical. It is related to why we have plotted induced charges against polarization, not displacement field in Fig. 3a. If polarizability of the wavefunctions were to affect the interaction with the Pt atoms in the interface, Fig. 3a would not be linear.

Parenthetically, the constrained-force approach is not standard in VASP, as far as I'm aware (but I could be mistaken). At any rate, if the authors used their own modified version of the code, this should be stated and some text provided in the SI on how this is implemented.

Author Response: We have changed the methods section of main text to: “Periodic density functional theory (DFT) calculations were performed with a customized version of Vienna ab initio simulation package (VASP) 5.4.4 modified for constrained-forces calculations.” Also, we have added a section in the SI on how it was implemented, including the link to source code.

Response to Referee 2

Authors are to be complimented on an exceptionally well written exposition on the potential influences of ferroelectric polarization on adsorption and reaction at on a metal/ferroelectric heterostructure. The Introduction does a very nice job of placing the work in the context of the field. I have only a few minor comments for consideration in a revision (in no particular order):

Author Response: We thank the reviewer for their positive assessment on our work.

1. The inference from the calculations is that external polarization can induce different responses of reactant and transition states (Figure 4d). It would be helpful in the Conclusions to draw a more direct line between this observation and the concept of a "programmable" catalyst. Could the effect be leveraged statically? Or dynamically? Both? Some discussion would help clarify what is currently a disconnect between the DFT observations and catalytic implications. Relevant to title as well.

Author Response: The referee raises an important point. The word 'dynamic' may refer to different time scales in different contexts. Strictly speaking, all our calculations are performed in static conditions, taking advantage of the Born-Oppenheimer approximation assuming static atoms. In this sense, our results are applicable to static experiments. However, the time scale of atomic displacements are vibrations are typically of the order of 10^{-12} seconds since the typical phonon frequencies are in the terahertz range. As a result, any dynamic catalysis experiment, which at best would be working in kilohertz frequencies, is static from the point of view of this paper, and our results are equally applicable to it.

In other words, our results can be used dynamically using existing microkinetic models. Such models are beyond the scope of the current manuscript, but in order to clarify this point that the referee raised, we edited the conclusion section so that it now reads: "In this study, we provided a detailed discussion of the thin-film Pt catalyst on ferroelectric PbTiO_3 in the static limit of the crystal, which corresponds to frequencies lower than the typical lattice vibration frequencies in the terahertz range. ... This result, combined with emerging studies of dynamic microkinetic modeling will provide a mechanism for faster, dynamic surface catalysis."

2. Induced polarization comes with an energy cost. Can this be estimated, and what is its magnitude relative to a chemical energy?

Author Response: The referee raises an interesting point. Any ferroelectric-based device is limited by the energy cost of polarization switching, and there is an ongoing effort to reduce this energy cost in devices. (See, for example, [J. Scott, "Applications of Modern Ferroelectrics", *Science* **315**, 954 (2007)].) We can provide a rough estimate of an upper limit to the switching energy scale in our system from Fig. 2b. and Fig 4. Energy needed to switch polarization is 0.08 eV/f.u PbTiO_3 . For example, In a catalytic capacitor of 100 nm thick insulator, this energy corresponds to 10 eV per active site (top). (Note that this value is bound to be a big overestimation, since the actual switching process involves motion of domain walls, the energy of which cannot be calculated from first principles precisely.) The maximum amount of change in the interaction energy of adsorption site and adsorbate is 0.2 eV. While the overall energy input may be higher than the change in interaction energy it causes, it should be noted the small changes in the interaction energy can lead to large increase in turnover frequency if applied dynamically. We would like to reiterate that better materials with low switching barriers are an active topic of research, where first-principles materials design plays an important role as well [Li & Birol (2020) *npj Computational Materials*, 6(1), 168]. While we do not get into the discussion and discovery of such materials in the current

manuscript, future work will surely focus on this aspect as well.

3. Model assumes fixed epitaxy between Pt overlayer and support and (I believe) constant lateral directions. What is lateral strain on system, and is it sensitive to polarization? Asked another way, what is interface energy and how does it change with polarization? Is the epitaxy assumption reasonable?

Author Response: The lattice parameter a used for model system is 3.906 Å, which corresponds to zero strain for tetragonal ground state of PbTiO₃. Lattice parameter for ground state Pt is 3.948 Å, which corresponds to about 1.0 % compressive strain. This information has been added to SI.

Lateral strain of oxides is indeed sensitive to polarization to some extent. We calculate the lattice constant a of non-polar ground state of PbTiO₃ to be 3.943 Å, which also corresponds to about 1% strain. In general, higher polarization induces larger strain in the direction of polarization, and lower strain in other directions.

We have not calculated the interface energy, but the epitaxy assumption is valid based on experimental evidence of epitaxial growth of metals on (100)-oriented perovskites. (Francis et al. (2006) *Thin Solid Films*, 496(2), 317-325.)

4. Along same lines, is the proposed concept limited to a planar heterostructure geometry? Is it applicable to the supported metal nanoparticle motif more common in catalysis.

Author Response: The geometry is restricted to using thin electrodes as catalysts, as the changes in the metal-insulator interface should affect the surface where reaction occurs. But it is applicable to supported metal nanoparticles, as long as it satisfies the thin electrode conditions. A recent experimental work of Pt nanoparticles on monolayer graphene is an example of such geometry. (Onn, T. M. et al. (2022). *Journal of the American Chemical Society*, 144(48), 22113-22127.)

5. Figure 4 shows MEP is fixed vs polarization, yet comments on page 8 indicate that path changes at some polarization. Could authors comment further? As what appears to underlie the observations are differential influences on Pt atoms, one might well imagine changes in reaction pathway to be a significant and confounding effect working against the "programmable" catalyst.

Author Response: Reviewer is correct that the reaction pathway changes can work against programmable catalyst, and we expect all reaction paths to change under significant changes to catalyst surface. However, if applied dynamically, we can choose the range of polarization change that the system will undergo that there is no significant change in the reaction pathway. Thus, it is not a big concern for application.

6. Page 4, the n=3 and n=6 cases are inconsistent between the Figure and the narrative at the bottom of the page.

Author Response: We thank the reviewer for pointing out the error. It has been adjusted correspondingly.

7. Unless I missed it, Figures 1b and 1c were not referred to in the text.

Author Response: We have added reference to figures 1a and 1b in the introductions. Figure 1c is referred in 1st paragraph of section 2.1.

8. While motivation is presented as desire to increase TOF, selectivity and stability are often equally/greater importance in practice. Maybe tangential to the work here, but importance considerations in wrt the "programmable" concept.

Author Response: We agree with the reviewer. However, alternate pathways of methanol decomposition is not studied, to keep the scope of this study focused. Application of dynamic microkinetic models based on methods presented from this study is future work to be done, as mentioned in the response to the first comment of the second reviewer.

List of Revisions

Section 1.0: Introduction

p.1 Added reference to Fig. 1a in main text.

Section 2.1: Polarized structures

p.3 Added reference to Fig. 1b in main text.

p.3 Changed the reference for the constrained-forces method from “introduced by Fu & Bellaiche” to “introduced by Sai et al. and Fu & Bellaiche”. Also added earlier citation by Sai, Rabe and Vanderbilt related to constrained-forces method.

p.4 Fixed recurring typos related to number of unit cells of PbTiO_3 .

Section 2.3: Adsorption and reaction control

p.7 Updated data in Fig 4 panel a, b and c with those of higher precision. (adsorption convergence force criterion from maximum of $0.02 \text{ eV}/\text{\AA}$ to $0.015 \text{ eV}/\text{\AA}$, NEB convergence force criterion from maximum of $0.045 \text{ eV}/\text{\AA}$ to $0.03 \text{ eV}/\text{\AA}$) Also fixed minor typos.

p.8 Updated initial state, transition state and final state energies in main text with higher precision data. Added an explanation on how transition state energy increases again at highest polarizations observed ($0.91 \text{ C}/\text{m}^2$) but activation energy is still lowered.

Section 3.0: Conclusions

p.10 Added further explanation of the scope of this research in the static limit.

p.10 Added a citation of microkinetic modeling research of dynamic programmable catalysis, mentioning that our research combined with the emerging researches of kinetic model can provide the first-principles insights of dynamic catalysis.

Supplementary Note 1. Computational Methods

SI p.1 Specified ground-state lattice parameter of Pt and biaxial strain.

SI p.2 Updated convergence criterion forces.

Supplementary Note 2. Constrained-forces calculations

SI p.5 Added more explanation on implementation of constrained-forces methods, along with link to source code.

Reviewers' Comments:

Reviewer #1:

Remarks to the Author:

The authors have comprehensively addressed all of my comments/concerns and I recommend that the paper be accepted for publication. Kudos to the authors on a nice piece of work!

Reviewer #2:

Remarks to the Author:

Thank you to the authors for their thorough responses. Unless I misread the response, it appears that they have not incorporated corresponding comments into the revised manuscript itself. I expect many readers will have the same questions that I raised in my first review, and thus for the work to have the impact the authors surely desire, they need to address those questions in the manuscript and/or SI.

With regards my first question regarding the connection to "programmable catalysis", I would ask that they reconsider and elaborate on their response. "Programmable catalysis" are the first two words of the title, yet it remains unclear how exactly the results "will provide a mechanism for faster, dynamic surface catalysis." This question needs to be addressed more substantially in the body of the work, not just the conclusions. Otherwise, the title is gratuitous and should be changed to more correctly capture the content of the work.

Reviewer #1 (Remarks to the Author):

The authors have comprehensively addressed all of my comments/concerns and I recommend that the paper be accepted for publication. Kudos to the authors on a nice piece of work!

Authors' Response:

We thank the referee for the positive assessment of our work, and the time and effort spent to review it.

Reviewer #2 (Remarks to the Author):

Thank you to the authors for their thorough responses. Unless I misread the response, it appears that they have not incorporated corresponding comments into the revised manuscript itself. I expect many readers will have the same questions that I raised in my first review, and thus for the work to have the impact the authors surely desire, they need to address those questions in the manuscript and/or SI.

With regards my first question regarding the connection to "programmable catalysis", I would ask that they reconsider and elaborate on their response. "Programmable catalysis" are the first two words of the title, yet it remains unclear how exactly the results "will provide a mechanism for faster, dynamic surface catalysis." This question needs to be addressed more substantially in the body of the work, not just the conclusions. Otherwise, the title is gratuitous and should be changed to more correctly capture the content of the work.

Authors' Response:

We thank the referee for emphasizing these points. In the updated manuscript, we made multiple additions to incorporate some remarks about the discussions in our previous response. (Changes in the text are shown in red in the submitted manuscript.) For example, we now discuss the finite energy cost of switching the ferroelectric polarization, and refer to references discussing how it may be reduced in the following way:

“In passing, we note that the ferroelectric double-well energy landscape leads to a finite energy cost of flipping the polarization, which may be an important factor in practical applications. A rough estimate of this energy scale for bulk-like PbTiO_3 is ~ 0.1 eV per formula unit. Reducing the switching energy and coercive field (for example by strain engineering) is desirable not only for catalytic but also for other applications of ferroelectrics, and is an area of active research.^{47–49}”

We also added a brief discussion of why we are not considering other reaction mechanisms in this work but how they may be important. Most importantly, we added the following paragraph to page 8 of the main text to explain and discuss and refer to other papers about the word ‘programmable’:

“The changes in absorption and transition states energies reveal two important points. First, the magnitudes of changes at the catalyst surface are consistent with the requirements of programmable catalysts for dynamic rate control.¹⁰ Variation in the extent of PbTiO_3 polarization can be achieved via changing the materials properties by strain, doping, layering, etc. as has been commonly done in perovskite-related oxides, and the polarization can be switched via an external potential bias applied to Pt layers acting as electrodes,^{56–59} thereby configuring the metal catalyst electronic properties via an external input with time (i.e., a program). Dynamic controlled modulation of catalyst electronic state

via external bias has been demonstrated to achieve resonance conditions which exist beyond conventional static catalytic rates;^{11,13,60} this resonance mechanism occurs only for catalysts that can be externally controlled to modulate the catalytic surface energy with time.^{61,62} For the considered methanol decomposition system, variation of device (Pt/PbTiO₃/Pt) polarization yields significant change in binding energy of key reaction intermediates, indicating that metal-ferroelectric devices provide capability for transient modulation to control the reaction surface chemistry via input voltage programs.”

Reviewers' Comments:

Reviewer #2:

Remarks to the Author:

The authors have adequately addressed my prior comments. The work is much improved and a solid addition to Nature Comm.

Reviewer #2 (Remarks to the Author):

The authors have adequately addressed my prior comments. The work is much improved and a solid addition to Nature Comm.

Authors' response:

We thank both reviewers for the effort and constructive criticisms they provided.